# Collaborative agglomeration level and spatial correlation of intercity manufacturing industry: An empirical study based on the cities of the Yangtze River Delta

**Kai Zhu**[1,2]*, **Wanxiang Sun**[1]

**1** School of Design and Architecture, Zhejiang University of Technology, Hangzhou, Zhejiang, China,
**2** School of Architecture, Southeast University, Nanjing, Jiangsu, China

* zk15@zjut.edu.cn

**Data Availability Statement:** All relevant data are within the paper and its Supporting Information files.

## Abstract

Taking 27 cities in the Yangtze River Delta as an example, the time section data from 2009 to 2019 are selected, and the location entropy index and the modified E-G index are introduced to measure the collaborative agglomeration level of intercity manufacturing industry in the Yangtze River Delta. The spatial weight matrix is constructed based on the highway mileage between cities. Using Moran's index and Local Moran's index, this article analyzes the spatial correlation of the collaborative agglomeration level of intercity manufacturing industry in the Yangtze River Delta. The results show that: Firstly, the overall agglomeration degree of manufacturing industry of cities in the Yangtze River Delta shows a fluctuating downward trend. The agglomeration degree of manufacturing industry in Jiangsu and Zhejiang Provinces has decreased, that of most cities in Anhui Province has increased steadily. Secondly, the collaborative agglomeration level of manufacturing industry between Shanghai, Nanjing, Hangzhou and other cities in Jiangsu and Zhejiang Provinces showed a downward trend, while the collaborative agglomeration level with Anhui cities increased steadily. The collaborative agglomeration level of manufacturing industry between Wuxi, Suzhou, Nanjing and cities with obvious advantages in manufacturing industry agglomeration degree shows an upward trend, while the collaborative agglomeration level with other cities has declined. The collaborative agglomeration level of intercity manufacturing industry in Hefei has generally improved, especially the collaborative agglomeration quality and depth of manufacturing industry between Hefei and other cities in Anhui Province have significantly improved. Thirdly, the regions with high collaborative agglomeration level of intercity manufacturing industry in the Yangtze River Delta are still mainly concentrated in Jiangsu and Zhejiang Provinces. Although there has been a trend of westward transfer in the past decade, the overall change range is small, and the spatial correlation of the collaborative agglomeration level of intercity manufacturing industry shows a weakening trend as a whole.

**Funding:** The research is funded by National Natural Science Foundation of China, grant number 52378080; and Zhejiang Provincial Philosophy and Social Science Planning Project, grant number 22NDQN208YB; and First-class Core Course Construction Project in Zhejiang University of Technology. Role of Funder statement: The author, Zhu Kai, is the project leader responsible for conducting relevant research activities. The project focuses on the manufacturing industry and economic development of the Yangtze River Delta, and the results of this article are an integral part of the project results.

**Competing interests:** The authors have declared that no competing interests exist.

# 1. Introduction

## 1.1 Background

Building a new development pattern of mutual promotion between domestic and international dual circulation is a major judgment on China's current economic situation and policies, among which internal circulation is an important focus for promoting high-quality development of China's dual circulation. As an important part of internal circulation, the circulation in the manufacturing industry has become the foundation of urban and even regional economic development with the continuous promotion of re industrialization and the return of the manufacturing industry. At the same time, promoting its high-quality development is also an inherent requirement for building a modern economic system. With the successive implementation of a new round of national and regional development strategies, the free circulation and optimized allocation of various production factors across the country and even globally have significantly accelerated. The industrial connections between different regions are becoming increasingly close, and the aggregation effect of capital is gradually emerging. Integration, especially industrial integration, has become a top priority for regional future development.

The Yangtze River Delta, as the leading force driving China's economic development and reform and opening up, is an important model area for regional integration development and the main battlefield for promoting high-quality development of the manufacturing industry. With the deepening of supply side structural reform, the high-quality integrated development of the Yangtze River Delta is gradually shifting towards a pattern of coordinated, integrated, and balanced development. In the future, the development of regional intercity industries will shift from competition to cooperation, and from siphon to interaction. Comprehensively promoting the transformation and upgrading of the manufacturing industry, actively cultivating new driving forces and engines, striving to create characteristic and advantageous industries, and laying a collaborative development pattern of interactive and complementary regional industrial chains are not only urgent need to optimize the internal industrial structure of the Yangtze River Delta urban agglomeration, but also inevitable stage for the integration of the Yangtze River Delta region to move towards higher levels and quality [1]. In this context, as the core content of regional integration, the collaborative agglomeration of intercity manufacturing industry is both a trend and an inevitable requirement. Measuring the collaborative agglomeration level of intercity manufacturing industry in the process of regional integration in the Yangtze River Delta is of great practical significance for clarifying the development direction of the manufacturing industry and promoting high-quality development of the integration of the Yangtze River Delta.

## 1.2 Literature review

The concept of industrial collaborative agglomeration originated from abroad. Ellison and Glaeser defined the spatial agglomeration phenomenon of industries as "co-agglomeration" [2]. Since then, many scholars have continuously enriched the research on industrial collaborative agglomeration and achieved fruitful theoretical results. Marshall's externality theory has laid the foundation for the emergence of the theory of industrial collaborative agglomeration [3]. The "center-periphery" model constructed by Krugman and Paul incorporates spatial factors into relevant research, and further develops the relevant content of industrial collaborative agglomeration [4]. Venables constructed a vertical correlation model from the perspective of heterogeneity, and pointed out that upstream and downstream industries with forward and backward connections tend to be closer to each other in space [5]. Ellison et al. explored the

degree of collaborative agglomeration between different industries in the United States based on relevant data analysis, and validated three influencing factors proposed by Marshall: the connection between intermediate inputs and final product suppliers, shared labor markets, increased information exchange and innovation opportunities [6]. Kind et al. introduced tax competition into the CPVL model and found that tax and transportation costs have the same impact on industrial agglomeration [7]. Forslid and Midelfart introduced the CPVL model to optimize the collaborative agglomeration of upstream and downstream industries in countries where government departments receive high wage openness [8]. Suedekum introduced the housing product sector into the Krugman center periphery model to analyze the industrial agglomeration characteristics of the three sectors [9]. When Kolko and Neumark analyzed the phenomenon of internal agglomeration in the service industry, they found that the internal agglomeration in the service industry was generated through trade exchanges between industries and the Spillover effect of knowledge [10]. Koh and Riedel replaced vertical correlation with horizontal correlation and established a three sector model for the two countries. They also found that there is spatial synergistic agglomeration between the intermediate product industry and the final product industry [11].

At present, research on industrial collaborative agglomeration mainly focuses on the collaborative agglomeration between manufacturing industry and productive service industry, covering multiple aspects such as existence analysis, collaborative agglomeration effect analysis, and collaborative agglomeration mechanism analysis. In terms of existence analysis, Krugman found that a large number of manufacturing industries in the United States have a high degree of production agglomeration, which is not limited to high-tech industries [12]. Duranto and Overman took micro enterprises as the research object, and adopted the method of econometric analysis to analyze the existence of collaborative agglomeration between productive service industry and manufacturing industry. The research results show that Marshall externality, knowledge spillovers and direct trade relations, circular cumulative effects and policy intervention are the main reasons for industrial collaborative agglomeration [13]. In terms of collaborative agglomeration effect analysis, Lu Jianbao conducted a study on the synergistic effect of productive service industry based on manufacturing industry agglomeration, and measured the impact of manufacturing industry agglomeration on productive service industry agglomeration. He found that manufacturing industry has a agglomeration driving effect on productive service industry, and manufacturing industry agglomeration is the main factor leading to the agglomeration of productive service industry [14]. Sheng Feng discussed the impact of the spatial agglomeration of productive service industry on the upgrading of manufacturing industry and its spatial spillover effect. He found that the spatial agglomeration of productive service industry can drive the upgrading of manufacturing industry [15]. In addition, some scholars have also proposed that the collaborative agglomeration between manufacturing industry and productive service industry has a significant positive promoting effect on industrial structure upgrading [16], regional and urban economic growth [17], and the improvement of total factor productivity [18]. In terms of collaborative agglomeration mechanism analysis, Gallagher subdivided transportation costs into physical and information transportation costs when studying the formation mechanism of industrial collaborative agglomeration, and analyzed that the Marshall factor under heterogeneous transportation costs is the source of the formation of industrial collaborative agglomeration [19]. Gabe and Abel examined the key role of Marshall's third factor and found that professional labor with similar knowledge tend to collaborate more, and the importance of knowledge sharing caused by this collaboration was greater at the urban scale than at the state level [20]. Through theoretical model deduction, Yang Renfa proposed that the effect of industrial collaborative agglomeration on environmental pollution is achieved through enterprise scale, technological progress, and economic

structure [21]. Zhang Zhidong and Qin Shuyue believed that the role of industrial agglomeration in green development is achieved through the dual effects of scale effect and crowding effect [22]. Chen Xi et al. proposed that institutions and policies, information transmission capabilities, economic development levels, manufacturing industry labor supply and transportation facility level are the main factors affecting the regional differences in collaborative agglomeration among manufacturing industry in China [23].

Overall, there are still some shortcomings in existing research, mainly including three aspects. Firstly, in terms of research perspective, industrial collaborative agglomeration refers to the agglomeration of different industries in a specific space, with a dual attribute of "industry-space". However, some studies have focused more on the single attribute of industry and explored the phenomenon of industrial collaborative agglomeration from an economic perspective, and neglected the importance of spatial attributes [24, 25]. Secondly, in terms of research objects, current research on the measurement of collaborative agglomeration level is mostly based on regions or provinces, and research focusing on the measurement of collaborative agglomeration level of intercity manufacturing industry is still rare [26, 27]. Thirdly, in terms of research content, existing research has mostly focused on the collaborative agglomeration between manufacturing industry and productive service industry or other industries, and there is relatively little research on the collaborative agglomeration between individual industries in the manufacturing industry [28, 29]. However, as an important carrier of regional integration in the Yangtze River Delta, the collaborative agglomeration of manufacturing industry between cities is also a focus of integration attention. Measuring the collaborative agglomeration level of intercity manufacturing industry will help better promote the process of regional integration in the Yangtze River Delta.

## 2. Objects and methods

### 2.1 Research objects and research ideas

**2.1.1 Research objects.** The Yangtze River Delta is located in the lower reaches of the Yangtze River in China, bordering on the Yellow Sea and the East China Sea, including Shanghai, Jiangsu, Zhejiang and Anhui Provinces. The manufacturing industry in the Yangtze River Delta has a strong foundation, which can be a leading and demonstration area for integrated development. Measuring the collaborative agglomeration level of intercity manufacturing industry has a certain typicality and representation. This article selects 27 cities in the Yangtze River Delta as the research objects, including Shanghai, Nanjing, Wuxi, Changzhou, Suzhou, Nantong, Yangzhou, Zhenjiang, Yancheng, Taizhou, Hangzhou, Ningbo, Wenzhou, Huzhou, Jiaxing, Shaoxing, Jinhua, Zhoushan, Taizhou, Hefei, Wuhu, Ma'anshan, Tongling, Anqing, Chuzhou, Chizhou, and Xuancheng. Focusing on the manufacturing industry, the sample study spans from 2009 to 2019, with data mainly involved the number of employees at the end of the year. These data are sourced from statistical yearbooks and statistical bureau websites of various provinces and cities, as well as China's urban statistical yearbooks. In addition, this article also uses the highway mileage data between cities, which are sourced from the cityscape website.

**2.1.2 Research ideas.** Building an integrated factor market is the key to regional integration. Generally speaking, a large economic volume brings about the agglomeration of production factors, which in turn further leads to the coordinated development of industries. With the deepening of integration, the flow and allocation of various production factors between regions are constantly accelerating, breaking the inherent understanding that economic volume determines factor agglomeration. The relationship between cities is becoming increasingly close, and the coordinated development of industries is no longer only influenced by urban economic volume, but also by the relationship between cities. The main idea of this

research is to measure the agglomeration level of manufacturing industry and the collaboration level based on agglomeration level, and then analyze the spatial synergy relationship of collaboration level.

Starting from the fact of agglomeration in the intercity manufacturing industry of the Yangtze River Delta, this article uses the industrial agglomeration index to measure the agglomeration level of manufacturing industry of 27 cities in the Yangtze River Delta to analyzes the differences in manufacturing industry agglomeration among different cities, and then uses the industrial collaborative agglomeration index to measure whether the agglomeration level of manufacturing industry of different cities are coordinated. Collaborative agglomeration of manufacturing industry is an advanced stage of manufacturing industry agglomeration, which not only reflects the close technological and economic connections between different cities, but also serves as an important basis for predicting the future development of intercity manufacturing industry in the Yangtze River Delta. In order to further explore the relationship between collaborative agglomeration and spatial distance of the intercity manufacturing industry in the Yangtze River Delta, that is, whether manufacturing industry collaborative agglomeration is short-distance or long-distance collaboration, this article uses Moran's I and Local Moran's I to conduct spatial correlation analysis on the collaborative agglomeration level of intercity manufacturing industry in the Yangtze River Delta to test whether the collaborative agglomeration level of intercity manufacturing industry is influenced by the proximity of urban spatial distances, and to provide more scientific and practical guidance for the development direction of the manufacturing industry in the Yangtze River Delta.

## 2.2 Research methods and indicator selection

**2.2.1 Measurement of collaborative agglomeration level.** *(1) Industrial agglomeration index*. Location entropy is an indicator that measures the degree of specialization of a specific industry in a certain region within the entire region. It is usually used to represent the balanced development of industries between cities or the industrial agglomeration level. It can eliminate the impact of regional scale differences and more accurately reflect the spatial distribution of industrial agglomeration. Therefore, this article uses the location entropy index to calculate the agglomeration level of manufacturing industry of cities in the Yangtze River Delta. The specific calculation formula is as follows:

$$LQ_{kj} = \frac{q_{kj}/q_j}{q_k/q} \tag{1}$$

where $LQ_{kj}$ and $q_{kj}$ respectively represent the location entropy and the relevant indicators of the k industry in j city; $q_j$ represents the relevant indicators of all industries in j city; $q_k$ represents the relevant indicators of the k industry in the entire region; and $q$ represents the relevant indicators of all industries in the entire region. In order to better reflect the industrial agglomeration degree and eliminate the impact of economic development level and price difference of the year, this article uses the number of urban unit employees at the end of the period as the main indicator to measure industrial agglomeration. Then, $q_{kj}$ represents the number of urban unit employees in the manufacturing industry of j city at the end of the period; $q_j$ represents the number of urban unit employees in all industries of j city at the end of the period; $q_k$ represents the number of urban unit employees in the manufacturing industry within the entire region at the end of the period; and q represents the number of urban unit employees in all industries within the entire region at the end of the period. The larger the value of $LQ_{kj}$, the higher the level of manufacturing industry agglomeration in the city. Generally speaking,

when $LQ_{kj} > 1$, it indicates that the k industry in j city has advantages throughout the entire region, which can be considered that the k industry is clustered in j city. When $LQ_{kj} < 1$, it indicates that the k industry in j city has disadvantages throughout the entire region.

*(2) Industrial collaborative agglomeration index.* The E-G index constructed by Ellison and Glaeser measures the strength of the relationship between enterprises' "co-location" according to the relevance of enterprise location, and weed out the market concentration ratio to eliminate the impact of enterprise size differences on industrial agglomeration, so as to more accurately measure the agglomeration degree of industrial geographical distribution [2]. To solve the problem of data acquisition and better adapt to the development of China's manufacturing industry, this article uses the revised E-G index to calculate the collaborative agglomeration level of intercity manufacturing industry in the Yangtze River Delta through the relative differences in industrial agglomeration. The calculation formula is as follows:

$$CO = 1 - |LQ_{ki} - LQ_{kj}|/\left(LQ_{ki} + LQ_{kj}\right) \tag{2}$$

where $LQ_{ki}$ and $LQ_{kj}$ respectively represent the agglomeration level of k industry in i city and j city, which are generally measured by location entropy.

This formula mainly reflects the quality of industrial collaborative agglomeration, without considering the depth of industrial collaborative agglomeration. Therefore, referring to the research of Chen Jianjun et al. [30] and Zhang Hu et al. [31], the above formula is further improved by considering both the quality and depth of industrial collaborative agglomeration, in order to better reflect the overall collaborative agglomeration level of intercity manufacturing industry in the Yangtze River Delta. The calculation formula is as follows:

$$COO = 1 - |LQ_{ki} - LQ_{kj}|/\left(LQ_{ki} + LQ_{kj}\right) + |LQ_{ki} - LQ_{kj}| \tag{3}$$

**2.2.2 Spatial correlation analysis.** *(1) Spatial weight matrix.* The establishment of spatial weight matrix is the foundation for conducting regional spatial correlation analysis and an important tool for reflecting regional spatial relationships [32, 33]. Commonly used spatial weight matrices include adjacent spatial weight matrix, geographical metric space weight matrix, economic metric space weight matrix, etc. The geographical metric space weight matrix regards geographical distance as an important factor in the measurement of interaction effect between space units, that is, a specific spatial unit will not only have interactive effects on its surrounding spatial units, but also will have interactive effects on non adjacent spatial units. The strength of this interactive influence is influenced by the geographical distance between spatial units. The geographical metric space weight matrix can more comprehensively and actually reflect the interaction effect between spatial units. Therefore, this article selects the reciprocal of the distance between cities in the Yangtze River Delta as the constituent element of the spatial matrix, and its specific expression is as follows:

$$W_{ij} = \begin{cases} \dfrac{1}{d_{ij}} & i \neq j \\ 0 & i = j \end{cases} \tag{4}$$

To ensure the consistency of calculation standards and the availability of data, this article uses the mileage of highways between cities as the measure of spatial distance between cities. The value of $d_{ij}$ is the mileage of highways between cities. The larger the mileage of highways between cities, the smaller the weight, and vice versa.

*(2) Measurement of spatial correlation.* Spatial auto-correlation is an important basis for testing whether a certain attribute in different geographical regions has spatial correlation features [34]. There are two main methods to measure the spatial correlation of variables: Moran's I and Local Moran's I. Among them, Moran's I is mainly used to reflect the spatial correlation characteristics of all spatial units, while Local Moran's I can reflect the characteristics of spatial heterogeneity between spatial units. The specific calculation formulas are as follows:

$$Moran's\ I = \frac{\sum_{i=1}^{n}\sum_{j=1}^{n} w_{ij}(x_i - \bar{x})(x_j - \bar{x})}{S^2 \sum_{i=1}^{n}\sum_{j=1}^{n} w_{ij}} \tag{5}$$

$$Local\ Moran's\ I = \frac{(x_i - \bar{x})}{S^2}\sum_{j=1}^{n} w_{ij}(x_i - \bar{x}) \tag{6}$$

where n is the number of cities; $x_i$ and $x_j$ respectively represent the observations of i city and j city; $w_{ij}$ represents spatial weight matrix; $S^2 = \frac{\sum_{i=1}^{n}(x_i - x)^2}{n}$ is the sample variance; and $\sum_{i=1}^{n}\sum_{j=1}^{n} w_{ij}$ is the sum of all spatial weights. The Moran's I has values between [–1,1]. When Moran's I>0, it indicates the existence of spatial positive correlation; when Moran's I<0, it indicates the existence of spatial negative correlation; and when Moran's I = 0, it indicates the absence of spatial correlation. The closer the value of Moran's I is to 1, the stronger the spatial positive correlation of the collaborative agglomeration level of intercity manufacturing industry. The closer the value of Moran's I is to -1, the stronger the spatial negative correlation of the collaborative agglomeration level of intercity manufacturing industry. The values of the Local Moran's I are also between [–1,1]. When Local Moran's I>0, it indicates that the observed values between two cities exhibit a co-directional correlation of "high -high" or "low-low" clustering. When Local Moran's I<0, it indicates that the observed values between two cities exhibit a reverse correlation of "high-low" or "low-high" clustering. When Local Moran's I = 0, Indicates that there is no spatial correlation between the observed values between two cities. The closer the absolute value of Local Moran's I is to 1, the more obvious the spatial correlation characteristics between cities.

## 3. Measurement of the collaborative agglomeration level of manufacturing industry between cities in the Yangtze River Delta

### 3.1 Agglomeration level of manufacturing industry

Using the number of urban unit employees at the end of the period as the main indicator, the agglomeration level of manufacturing industry of cities in the Yangtze River Delta was measured, and the results are shown in Table 1. From the average agglomeration level of manufacturing industry of various cities, the agglomeration degree of manufacturing industry in Suzhou, Jiaxing, Wuxi, Changzhou, and Ningbo all exceeds 4.00, which is much higher than other cities. The development of manufacturing industry in these cities started early, with a relatively strong foundation, and the agglomeration of manufacturing industry has formed a certain scale. The agglomeration degree of manufacturing industry in Jinhua, Zhoushan, Anqing, and Chizhou is still around 1, which is far lower than the average level of cities in the Yangtze River Delta. The analysis may be related to their urban development level and leading industries, and their manufacturing industry foundation is relatively weak and has not yet formed a agglomeration scale.

**Table 1. The manufacturing industry agglomeration degree of cities in the Yangtze River Delta.**

| Region/City | 2009 | 2010 | 2011 | 2012 | 2013 | 2014 | 2015 | 2016 | 2017 | 2018 | 2019 | Average |
|---|---|---|---|---|---|---|---|---|---|---|---|---|
| Yangtze River Delta | 3.90 | 3.89 | 3.88 | 3.30 | 2.60 | 2.58 | 2.68 | 2.84 | 3.00 | 3.18 | 3.38 | 3.20 |
| Shanghai | 3.48 | 3.43 | 3.65 | 3.43 | 2.49 | 2.21 | 2.26 | 2.44 | 2.49 | 2.62 | 2.31 | 2.80 |
| Nanjing | 3.54 | 3.60 | 3.55 | 3.16 | 1.97 | 1.82 | 1.98 | 2.00 | 2.03 | 2.11 | 2.29 | 2.55 |
| Wuxi | 5.10 | 5.22 | 5.22 | 4.68 | 4.17 | 4.30 | 4.49 | 4.67 | 5.11 | 5.61 | 5.84 | 4.95 |
| Changzhou | 4.33 | 4.22 | 3.98 | 3.99 | 3.28 | 3.39 | 3.79 | 4.05 | 4.39 | 4.64 | 5.21 | 4.12 |
| Suzhou | 6.48 | 6.56 | 5.94 | 5.75 | 4.94 | 5.22 | 5.51 | 5.89 | 6.28 | 6.92 | 7.20 | 6.06 |
| Nantong | 4.55 | 4.57 | 4.32 | 3.96 | 1.58 | 1.58 | 1.78 | 1.92 | 1.87 | 1.95 | 2.32 | 2.76 |
| Yangzhou | 3.24 | 3.28 | 3.36 | 3.14 | 1.83 | 2.05 | 2.11 | 2.22 | 2.28 | 2.33 | 2.23 | 2.55 |
| Zhenjiang | 4.55 | 4.70 | 4.44 | 4.23 | 3.38 | 3.75 | 3.95 | 4.22 | 3.91 | 3.84 | 3.70 | 4.06 |
| Yancheng | 2.90 | 3.01 | 3.00 | 2.74 | 2.22 | 2.24 | 2.23 | 2.33 | 2.32 | 2.18 | 2.48 | 2.51 |
| Taizhou | 3.84 | 3.93 | 3.85 | 3.46 | 2.12 | 2.06 | 2.20 | 2.23 | 2.34 | 2.02 | 2.36 | 2.76 |
| Hangzhou | 3.34 | 3.11 | 3.37 | 2.48 | 1.87 | 1.81 | 1.87 | 1.90 | 2.10 | 2.30 | 2.40 | 2.41 |
| Ningbo | 4.28 | 4.33 | 4.87 | 4.09 | 3.34 | 3.34 | 3.46 | 3.57 | 3.81 | 4.42 | 4.67 | 4.02 |
| Wenzhou | 4.53 | 4.48 | 4.02 | 3.30 | 2.42 | 2.35 | 2.29 | 2.47 | 2.34 | 2.39 | 2.92 | 3.05 |
| Huzhou | 4.82 | 4.82 | 4.20 | 3.37 | 2.76 | 2.88 | 2.97 | 3.22 | 3.46 | 3.74 | 4.15 | 3.67 |
| Jiaxing | 6.24 | 6.19 | 5.95 | 5.38 | 4.45 | 4.52 | 4.55 | 4.80 | 5.10 | 5.73 | 6.34 | 5.39 |
| Shaoxing | 3.74 | 3.34 | 2.92 | 2.43 | 2.03 | 2.00 | 1.99 | 2.03 | 2.30 | 2.31 | 2.82 | 2.54 |
| Jinhua | 2.59 | 2.80 | 2.71 | 1.57 | 1.28 | 1.23 | 1.28 | 1.31 | 1.61 | 1.56 | 2.09 | 1.82 |
| Zhoushan | 2.58 | 2.50 | 2.25 | 2.00 | 1.66 | 1.69 | 1.75 | 1.84 | 1.59 | 1.75 | 2.05 | 1.97 |
| Taizhou | 3.03 | 3.13 | 3.50 | 3.00 | 2.58 | 2.50 | 2.40 | 2.79 | 2.97 | 3.37 | 3.99 | 3.02 |
| Hefei | 2.20 | 2.39 | 2.57 | 2.11 | 1.71 | 1.88 | 1.95 | 2.03 | 2.15 | 2.05 | 2.21 | 2.11 |
| Wuhu | 3.76 | 4.05 | 3.50 | 3.38 | 2.90 | 2.94 | 3.11 | 3.41 | 3.53 | 4.03 | 4.15 | 3.52 |
| Ma'anshan | 4.32 | 4.29 | 3.48 | 3.08 | 2.13 | 2.20 | 2.31 | 2.42 | 2.51 | 3.29 | 3.42 | 3.04 |
| Tongling | 4.13 | 4.09 | 4.53 | 4.03 | 3.05 | 3.09 | 2.86 | 3.04 | 3.24 | 3.45 | 3.87 | 3.58 |
| Anqin | 1.22 | 1.22 | 1.28 | 1.10 | 1.78 | 1.81 | 2.13 | 2.30 | 2.40 | 2.99 | 2.92 | 1.92 |
| Chuzhou | 1.81 | 1.70 | 1.79 | 1.75 | 2.13 | 2.13 | 2.48 | 2.69 | 3.22 | 3.55 | 4.15 | 2.49 |
| Chizhou | 1.31 | 1.41 | 1.43 | 1.23 | 1.34 | 1.51 | 1.55 | 1.76 | 1.96 | 2.35 | 2.34 | 1.65 |
| Xuancheng | 2.35 | 2.47 | 5.70 | 1.38 | 1.20 | 2.22 | 2.42 | 2.59 | 2.82 | 3.54 | 3.64 | 2.76 |

From the perspective of temporal changes, the overall manufacturing industry agglomeration degree of cities in the Yangtze River Delta shows a fluctuating downward trend, which may be related to the transformation and upgrading of their industrial structure and the development towards service-oriented direction. Among the seven cities in the Yangtze River Delta, namely Shanghai, Nanjing, Wuxi, Suzhou, Hangzhou, Ningbo, and Hefei, the manufacturing industry agglomeration of Shanghai, Nanjing, and Hangzhou has shown an overall downward trend, which may be related to the relocation of their manufacturing industry and the transformation towards service industry development. Suzhou, Wuxi, and Ningbo have shown an overall upward trend in manufacturing agglomeration in the past decade due to their strong manufacturing industry foundation, and the process of manufacturing industry upgrading is constantly advancing. Due to the continuous influx of manufacturing industry in recent years, the agglomeration degree of manufacturing industry in Hefei has begun to show an upward trend. Looking at the overall changes in various provinces and cities, the agglomeration degree of manufacturing industry in most cities in Jiangsu and Zhejiang Provinces has decreased, while in most cities in Anhui Province has steadily increased, and the phenomenon of manufacturing industry shifting westward is obvious (Fig 1).

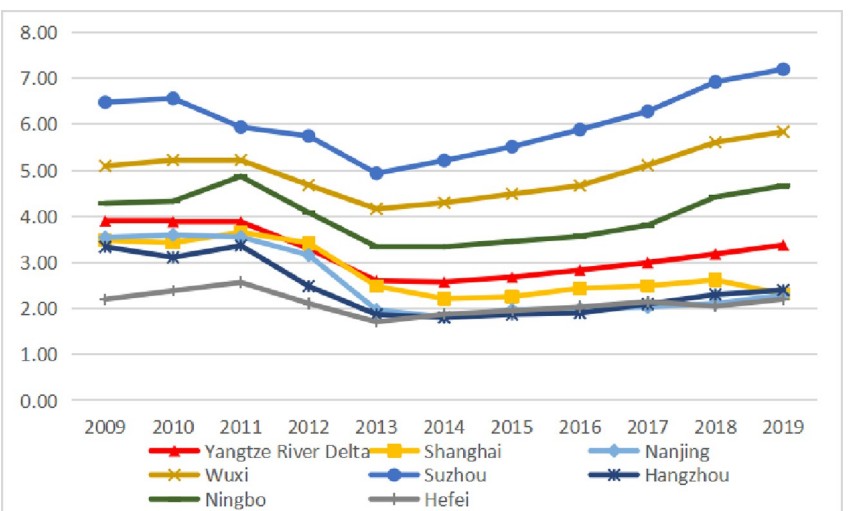

**Fig 1. Changes in manufacturing industry agglomeration of cities in the Yangtze River Delta from 2009 to 2019.**

## 3.2 Collaboration level of intercity manufacturing industry

On the basis of measuring the agglomeration level of manufacturing industry in various cities, in order to further explore whether the agglomeration level of manufacturing industry among cities is coordinated, this article uses formula (3) to calculate the collaboration level of intercity manufacturing industry in the Yangtze River Delta in 2009, 2012, 2015, and 2019.

The analysis results indicate that the collaboration level of intercity manufacturing industry between cities such as Suzhou, Jiaxing, Wuxi, Ningbo, and other cities is relatively high, while the collaboration level of intercity manufacturing industry between cities in Anhui Province, northern Jiangsu Province, southern Zhejiang Province, and other cities is relatively low, showing an overall trend of weakening from the center to the periphery. In terms of specific measurement values, the average collaboration level of intercity manufacturing industry between cities such as Suzhou-Jiaxing, Wuxi-Suzhou, Wuxi-Jiaxing, Suzhou-Changzhou, Suzhou-Zhenjiang, Suzhou-Ningbo, Suzhou-Huzhou, Suzhou-Tongling, Suzhou-Wuhu, Ningbo-Jiaxing, etc. all exceeds 10.00. Both the quality and depth of collaborative agglomeration of manufacturing industry have obvious advantages. The analysis found that these cities have a relatively high degree of manufacturing industry agglomeration and small intercity differences, resulting in a relatively high collaboration level. The collaboration level of intercity manufacturing industry between cities such as Yangzhou-Chizhou, Yancheng-Chizhou, Hangzhou-Chizhou, Hangzhou-Xuancheng, Shaoxing-Chizhou, Jinhua-Zhoushan, Hefei-Anqing, and Hefei-Chizhou is still below 5.00. The analysis may be due to the presence of cities with relatively low manufacturing industry agglomeration, resulting in significant differences in manufacturing industry agglomeration between cities. Therefore, there is significant room for improvement in the quality and depth of collaborative agglomeration of manufacturing industry among these cities.

From the perspective of temporal changes, the trend of manufacturing industry transfer from Jiangsu and Zhejiang Provinces to Anhui Province is obvious, and the collaborative agglomeration degree of intercity manufacturing industry between cities in Anhui Province is constantly strengthening. In terms of specific cities, the collaborative agglomeration level of intercity manufacturing industry between Shanghai, Nanjing, Hangzhou and other cities in Jiangsu and Zhejiang Provinces has shown a downward trend in the past decade, while the

collaborative agglomeration level of intercity manufacturing industry between cities in Anhui Province has steadily improved. The collaborative agglomeration level of intercity manufacturing industry between Wuxi, Suzhou, Ningbo and cities with obvious advantages in manufacturing industry agglomeration has shown an upward trend in the past decade, while the collaborative agglomeration level of intercity manufacturing industry with other cities has also decreased. The collaborative agglomeration level of intercity manufacturing industry in Hefei has generally improved in the past decade, especially with the significant improvement in the collaborative agglomeration quality and depth of intercity manufacturing industry with various cities in Anhui Province. Overall, the regions with high collaborative agglomeration level of intercity manufacturing industry in the Yangtze River Delta are still mainly concentrated in Jiangsu and Zhejiang Provinces. Although there has been a trend of westward transfer in the past decade, the overall change is not significant. It can be found that the trend of the collaborative agglomeration level of intercity manufacturing industry is related to the trend of manufacturing industry agglomeration in each city itself. Generally speaking, the collaborative agglomeration level of intercity manufacturing industry between cities with high manufacturing industry agglomeration is also relatively high, and the collaborative agglomeration level of intercity manufacturing industry rises and falls with the rises and falls of manufacturing industry agglomeration.

## 4. Spatial correlation analysis of the collaborative agglomeration level of manufacturing industry between cities in the Yangtze River Delta

From the above calculation results, it can be seen that the collaborative agglomeration level of intercity manufacturing industry in the Yangtze River Delta shows a non equilibrium feature, with both short distance and long distance collaborative phenomena occurring simultaneously. To explore whether the collaborative agglomeration characteristics of manufacturing industry between different cities have spatial dependence, this article uses the Moran's I and Local Moran's I indices to conduct spatial correlation analysis on the collaborative agglomeration level of intercity manufacturing industry in the Yangtze River Delta, in order to test the mutual relationship between the collaborative agglomeration level of intercity manufacturing industry and spatial distance.

### 4.1 Spatial auto-correlation test

In order to test the overall spatial correlation of collaborative agglomeration level of intercity manufacturing industry in the Yangtze River Delta, according to the above measured collaborative agglomeration level of intercity manufacturing industry, the overall collaborative agglomeration level in each city is expressed by summing up the values between each city and 26 other cities. Then, Stata software is used to calculate the Moran's I of the collaborative agglomeration level of intercity manufacturing industry in the Yangtze River Delta after standardizing the spatial weight matrix by using the geographical metric space weight matrix. The calculation results are shown in Table 2.

As shown in the table, the Moran's I distribution of the collaborative agglomeration level of intercity manufacturing industry in the Yangtze River Delta from 2009 to 2019 ranges from 0.01 to 0.08, with values greater than 0, indicating a spatial positive correlation between the collaborative agglomeration level of intercity manufacturing industry. In terms of significance, the p-values of the collaborative agglomeration level of intercity manufacturing industry from 2009 to 2019 were all less than 0.1, passing the 10% significance test, indicating that there is a significant spatial positive correlation in the collaborative agglomeration level of intercity

**Table 2. Moran' I index of collaborative agglomeration level of intercity manufacturing industry in the Yangtze River Delta from 2009 to 2019.**

| Year | Moran's I | Sd (I) | Z value | P value |
|---|---|---|---|---|
| 2009 | 0.083 | 0.033 | 3.719 | 0.000 |
| 2010 | 0.074 | 0.032 | 3.468 | 0.000 |
| 2011 | 0.041 | 0.033 | 2.441 | 0.007 |
| 2012 | 0.070 | 0.033 | 3.318 | 0.000 |
| 2013 | 0.025 | 0.032 | 1.970 | 0.024 |
| 2014 | 0.021 | 0.032 | 1.859 | 0.032 |
| 2015 | 0.030 | 0.032 | 2.148 | 0.016 |
| 2016 | 0.025 | 0.032 | 1.990 | 0.023 |
| 2017 | 0.025 | 0.032 | 1.990 | 0.023 |
| 2018 | 0.013 | 0.032 | 1.593 | 0.056 |
| 2019 | 0.009 | 0.032 | 1.469 | 0.071 |

manufacturing industry in the Yangtze River Delta, reflecting significant spatial agglomeration characteristics. From the perspective of temporal changes, the overall spatial correlation shows a weakening trend, indicating that the impact of spatial distance on the collaborative agglomeration of intercity manufacturing industry is continuously decreasing, and the cross city collaborative cooperation of manufacturing industry is gradually breaking through the limitations of spatial distance.

## 4.2 Local spatial correlation analysis

Due to the inability of the Moran's I index to reflect the spatial clustering characteristics and spatial dependence of the collaborative agglomeration level of intercity manufacturing industry, this article further studies the local spatial correlation of the collaborative agglomeration level of intercity manufacturing industry in the Yangtze River Delta by drawing a Moran scatter plot. This article selects Moran scatter plots for comparative analysis of the collaborative agglomeration level of intercity manufacturing industry in four years of 2009, 2012, 2015, and 2019, as shown in Fig 2.

Fig 2 shows the spatial distribution pattern and dynamic evolution of the collaborative agglomeration level of intercity manufacturing industry in the Yangtze River Delta, further reflecting the spatial correlation and heterogeneity characteristics of the collaborative agglomeration level of manufacturing industry among cities. From the graph, it can be seen that some cities located in the first quadrant, which is known as the "high-high" clustering, are mainly clustered in Jiangsu and Zhejiang Provinces, while some cities located in the third quadrant, which is known as the "low-low" clustering, are mainly clustered in Anhui Province. This indicates that there is a local high value clustering feature in the collaborative agglomeration level of manufacturing industry in some cities in Jiangsu and Zhejiang Provinces, while some cities in Anhui Province have a local low value clustering feature, which is a spatial positive correlation. Overall, the number of cities located in the first and third quadrants continues to decrease, while the number of cities located in the second quadrant, which is the "low-high" value cluster, and the number of cities located in the fourth quadrant, which is the "high-low" value cluster, continue to increase. These cities deviate from the global spatial positive correlation feature and exhibit local negative correlation. As time goes by, the scatter distribution in each quadrant shows a trend of shifting from relative diffusion to relative concentration, indicating that the spatial differences in the collaborative agglomeration level of manufacturing

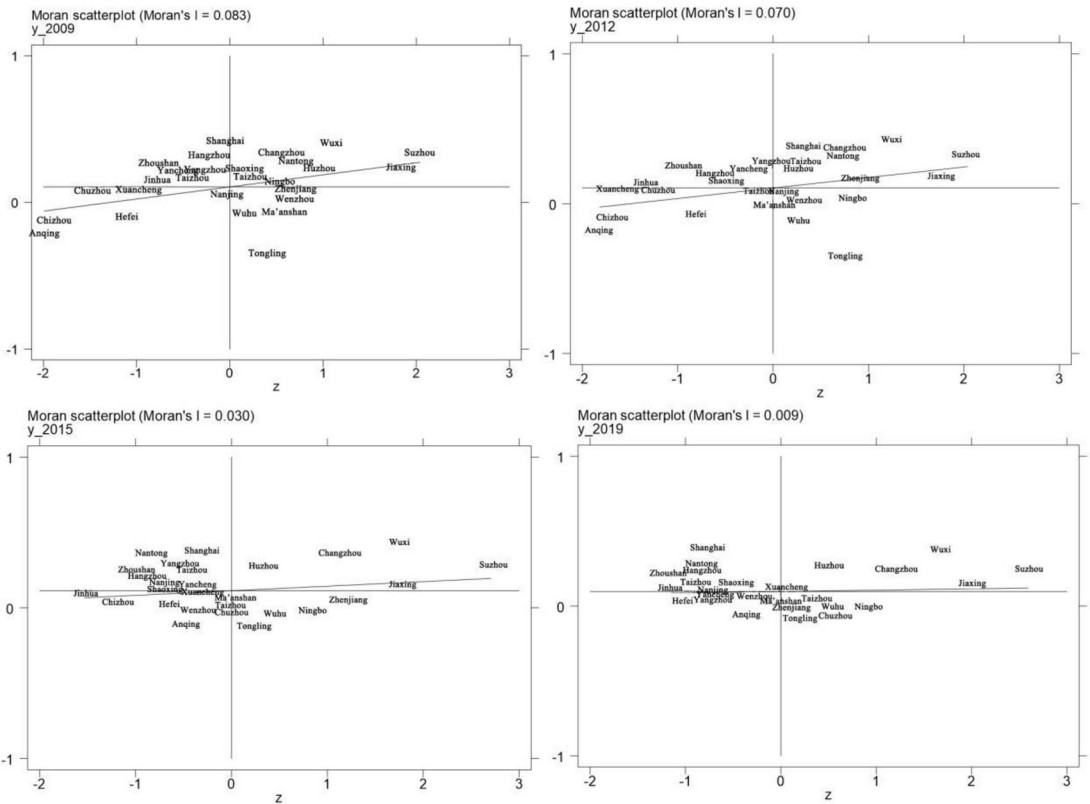

**Fig 2. Moran scatter diagram of the collaborative agglomeration level of intercity manufacturing industry in the Yangtze River Delta in 2009, 2012, 2015 and 2019.**

industry between cities are further narrowing, and spatial constraints are gradually being broken.

## 5. Conclusions and discussions

The collaborative agglomeration of intercity manufacturing industry is the core content of regional integration. This article uses the location entropy index and the modified E-G index to calculate the agglomeration level of manufacturing industry and collaboration level of intercity manufacturing industry of cities in the Yangtze River Delta. The results show that the manufacturing industry of cities in the Yangtze River Delta is relatively concentrated. However, the overall agglomeration level is showing a downward trend, with significant intercity differences in the collaborative agglomeration level of manufacturing industry, and a clear trend of transfer to Anhui Province. The spatial correlation analysis of the collaborative agglomeration level of intercity manufacturing industry in the Yangtze River Delta was conducted using the Moran's I and Local Moran's I indices. It can be seen that the collaborative agglomeration level of intercity manufacturing industry in the Yangtze River Delta has a significant spatial positive correlation, but the overall correlation shows a weakening trend. The spatial differences in the collaborative agglomeration level of manufacturing industry between cities are constantly narrowing.

With the transformation and upgrading of the manufacturing industry internally and the external transfer and evacuation, the flow of factors, technology diffusion, and information exchange between cities have influenced the collaborative agglomeration level of

manufacturing industry between cities. The spatial differences in the collaborative agglomeration level of manufacturing industry between cities are constantly narrowing, and the trend of cross city and cross province collaborative cooperation in manufacturing industry is becoming increasingly evident. Breaking down the limitations of spatial distance and administrative barriers, promoting the collaborative agglomeration of intercity manufacturing industry, and accelerating the process of regional integration can be approached from the following three aspects:

Firstly, using urban agglomerations as carriers, strengthen the specialized division of labor and cooperation between cities, and enhance the ability of small and medium-sized cities to undertake industrial transfer. For cities with high collaborative agglomeration level, it is necessary to accelerate the transformation and upgrading of the manufacturing industry, further strengthen the advantages of manufacturing industry agglomeration, and strengthen and optimize the manufacturing industry.

Secondly, focus on differentiated development of the manufacturing industry and reduce regional homogeneous competition. Based on the resource endowments and comparative advantages of each city within the urban cluster, it is necessary to clarify their respective manufacturing development priorities combined with the foundation of manufacturing industry development.

Thirdly, break down administrative barriers and spatial boundaries, and strengthen cooperation between governments. It is necessary to strengthen the division of labor and cooperation in the manufacturing industry among cities within the urban agglomeration and improve the integration capacity of industrial resources in the urban agglomeration, so as to promote the free flow of production factors within the urban agglomeration.

This article aims to measure the collaborative agglomeration level of manufacturing industry of cities in the Yangtze River Delta. The conclusions drawn from the relevant calculation results and the discussions extended provide certain policy guidance for the differentiated development and collaborative agglomeration of urban manufacturing industry.

## Supporting information

**S1 Dataset.**
(XLSX)

## Author Contributions

**Conceptualization:** Kai Zhu.

**Data curation:** Wanxiang Sun.

**Formal analysis:** Wanxiang Sun.

**Investigation:** Wanxiang Sun.

**Supervision:** Kai Zhu.

**Writing – original draft:** Wanxiang Sun.

**Writing – review & editing:** Kai Zhu.

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
