## [Decision Letter · Decision Letter 0]

29 Aug 2023

PONE-D-23-25180Research on the Collaborative Agglomeration Level and Spatial Correlation of Intercity Manufacturing Industry: An Empirical Study Based on the Central Cities of the Yangtze River DeltaPLOS ONE

Dear Dr. kai,

Thank you for submitting your manuscript to PLOS ONE. After careful consideration, we feel that it has merit but does not fully meet PLOS ONE’s publication criteria as it currently stands. Therefore, we invite you to submit a revised version of the manuscript that addresses the points raised during the review process.

We look forward to receiving your revised manuscript.

Kind regards,

Liang Zhuang, Ph.D.

Academic Editor

PLOS ONE

Journal Requirements:

2.In your Data Availability statement, you have not specified where the minimal data set underlying the results described in your manuscript can be found. PLOS defines a study's minimal data set as the underlying data used to reach the conclusions drawn in the manuscript and any additional data required to replicate the reported study findings in their entirety. All PLOS journals require that the minimal data set be made fully available. For more information about our data policy, please see http://journals.plos.org/plosone/s/data-availability.

3. We note that Figure 2 in your submission contain map/satellite images which may be copyrighted. All PLOS content is published under the Creative Commons Attribution License (CC BY 4.0), which means that the manuscript, images, and Supporting Information files will be freely available online, and any third party is permitted to access, download, copy, distribute, and use these materials in any way, even commercially, with proper attribution. For these reasons, we cannot publish previously copyrighted maps or satellite images created using proprietary data, such as Google software (Google Maps, Street View, and Earth). For more information, see our copyright guidelines: http://journals.plos.org/plosone/s/licenses-and-copyright.

Reviewers' comments:

Reviewer's Responses to Questions

**Comments to the Author**

1. Is the manuscript technically sound, and do the data support the conclusions?

Reviewer #1: Yes

Reviewer #2: Yes

2. Has the statistical analysis been performed appropriately and rigorously? 

Reviewer #1: Yes

Reviewer #2: Yes

3. Have the authors made all data underlying the findings in their manuscript fully available?

Reviewer #1: No

Reviewer #2: Yes

4. Is the manuscript presented in an intelligible fashion and written in standard English?

Reviewer #1: No

Reviewer #2: Yes

5. Review Comments to the Author

Reviewer #1: This is an original research paper on the collaborative agglomeration level and spatial correlation of manufacturing industry in the cities of the Yangtze River Delta. Using the quantative methods and GIS analysis to show the temporal changes of collaborative agglomeration level and spatial correlation of manufacturing industry in the cities of the Yangtze River Delta, the results are credible and indicating the trend of the spatial changes of manufaturing industry in the YRD.

Reviewer #2: This is an interesting paper that discusses the collaborative agglomeration level and spatial correlation of intercity manufacturing industry of China's Yangtze River Delta. The paper provides a new perspective to demonstrate intercity collaboration, which is of great significance to understanding the current process of regional integration of the Yangtze River Delta. I would like to suggest it to be published in this journal after addressing the following small issues.

1. A figure of the research objects should be provided in section 2.1.1 so that people could have a sense of where the Yangtze River Delta is.

2. The data source needs to be explicitly described.

3. Texts in Figure 3 should be changed to English

4. Policy implications from empirical findings should be discussed at the end.

6. PLOS authors have the option to publish the peer review history of their article (what does this mean?). If published, this will include your full peer review and any attached files.

Reviewer #1: No

Reviewer #2: No

---

## [Author Response · Author response to Decision Letter 0]

25 Sep 2023

Dear reviewers, thank you very much for your review and valuable comments. We have revised the article according to your comments. The relevant modifications are briefly described as follows:

Point 1: A figure of the research objects should be provided in section 2.1.1 so that people could have a sense of where the Yangtze River Delta is. 

Response 1: Due to the fact that the location map of the Yangtze River Delta may involve maps/satellite images which may be copyrighted, we have provided a textual description of the approximate location of the Yangtze River Delta as an alternative in the text section. 

Point 2: The data source needs to be explicitly described. 

Response 2: In section 2.2.1, we provided a detailed description of the data sources.

Point 3: Texts in Figure 3 should be changed to English. 

Response 3: We have made modifications to Figure 3 by replacing Chinese with English.

Point 4: Policy implications from empirical findings should be discussed at the end. 

Response 4: In the conclusion and discussion section, we supplemented the policy implications from empirical findings.

---

## [Decision Letter · Decision Letter 1]

18 Oct 2023

Collaborative agglomeration level and spatial correlation of intercity manufacturing industry: An empirical study based on the cities of the Yangtze River Delta

PONE-D-23-25180R1

Dear Dr. kai,

We’re pleased to inform you that your manuscript has been judged scientifically suitable for publication and will be formally accepted for publication once it meets all outstanding technical requirements.

Kind regards,

Liang Zhuang, Ph.D.

Academic Editor

PLOS ONE

Additional Editor Comments (optional):

Reviewers' comments:

Reviewer's Responses to Questions

**Comments to the Author**

1. If the authors have adequately addressed your comments raised in a previous round of review and you feel that this manuscript is now acceptable for publication, you may indicate that here to bypass the “Comments to the Author” section, enter your conflict of interest statement in the “Confidential to Editor” section, and submit your "Accept" recommendation.

Reviewer #2: All comments have been addressed

Reviewer #3: All comments have been addressed

2. Is the manuscript technically sound, and do the data support the conclusions?

Reviewer #2: Yes

Reviewer #3: Yes

3. Has the statistical analysis been performed appropriately and rigorously? 

Reviewer #2: Yes

Reviewer #3: Yes

4. Have the authors made all data underlying the findings in their manuscript fully available?

Reviewer #2: Yes

Reviewer #3: Yes

5. Is the manuscript presented in an intelligible fashion and written in standard English?

Reviewer #2: Yes

Reviewer #3: Yes

6. Review Comments to the Author

Reviewer #2: (No Response)

Reviewer #3: The authors have fully revised the paper in response to previous reviews. However, to ensure the overall quality of the paper, the paper still needs to be improved. First, the location map is still needed in this paper, and the authors can choose the official map to make the location map. Second, the current policy implication needs to be further discussed. Finally, the formats of this paper need to be adjusted and unified, such as the texts, formulas (italic or non-italic), tables (three-line table), figures and references and so on.

7. PLOS authors have the option to publish the peer review history of their article (what does this mean?). If published, this will include your full peer review and any attached files.

Reviewer #2: No

Reviewer #3: No

---

## [Editor Report · Acceptance letter]

3 Nov 2023

PONE-D-23-25180R1 

Collaborative agglomeration level and spatial correlation of intercity manufacturing industry: An empirical study based on the cities of the Yangtze River Delta 

Dear Dr. Zhu:

I'm pleased to inform you that your manuscript has been deemed suitable for publication in PLOS ONE. Congratulations! Your manuscript is now with our production department. 

Kind regards, 

on behalf of

Assistant Professor Liang Zhuang 

Academic Editor

PLOS ONE